# Replication of null results: Absence of evidence or evidence of absence?

**Samuel Pawel\*[†], Rachel Heyard[†], Charlotte Micheloud, Leonhard Held**

Epidemiology, Biostatistics and Prevention Institute, Center for Reproducible Science, University of Zurich, Zurich, Switzerland

**Abstract** In several large-scale replication projects, statistically non-significant results in both the original and the replication study have been interpreted as a 'replication success.' Here, we discuss the logical problems with this approach: Non-significance in both studies does not ensure that the studies provide evidence for the absence of an effect and 'replication success' can virtually always be achieved if the sample sizes are small enough. In addition, the relevant error rates are not controlled. We show how methods, such as equivalence testing and Bayes factors, can be used to adequately quantify the evidence for the absence of an effect and how they can be applied in the replication setting. Using data from the Reproducibility Project: Cancer Biology, the Experimental Philosophy Replicability Project, and the Reproducibility Project: Psychology we illustrate that many original and replication studies with 'null results' are in fact inconclusive. We conclude that it is important to also replicate studies with statistically non-significant results, but that they should be designed, analyzed, and interpreted appropriately.

**\*For correspondence:**
samuel.pawel@uzh.ch

[†]These authors contributed equally to this work

**Competing interest:** The authors declare that no competing interests exist.

## eLife assessment

This work provides a **valuable** contribution and assessment of what it means to replicate a null study finding, and what are the appropriate methods for doing so (apart from a rote p-value assessment). Through a **convincing** re-analysis of results from the Reproducibility Project: Cancer Biology using frequentist equivalence testing and Bayes factors, the authors demonstrate that even when reducing 'replicability success' to a single criterion, how precisely replication is measured may yield differing results. Less focus is directed to appropriate replication of non-null findings.

## Introduction

*Absence of evidence is not evidence of absence* – the title of the 1995 paper by Douglas Altman and Martin Bland has since become a mantra in the statistical and medical literature (*Altman and Bland, 1995*). Yet, the misconception that a statistically non-significant result indicates evidence for the absence of an effect is unfortunately still widespread (*Greenland, 2011*; *Makin and Orban de Xivry, 2019*). Such a 'null result' – typically characterized by a *p*-value $p > 0.05$ for the null hypothesis of an absent effect – may also occur if an effect is actually present. For example, if the sample size of a study is chosen to detect an assumed effect with a power of 80%, null results will incorrectly occur 20% of the time when the assumed effect is actually present. If the power of the study is lower, null results will occur more often. In general, the lower the power of a study, the greater the ambiguity of a null result. To put a null result in context, it is, therefore, critical to know whether the study was adequately powered and under what assumed effect the power was calculated (*Hoenig and Heisey, 2001*; *Greenland, 2012*). However, if the goal of a study is to explicitly quantify the evidence for the absence of an effect, more appropriate methods designed for this task, such as equivalence testing

(*Wellek, 2010*; *Lakens, 2017*; *Senn, 2021*) or Bayes factors (*Kass and Raftery, 1995*; *Goodman, 1999*; *Goodman, 2005*; *Dienes, 2014*; *Keysers et al., 2020*), should be used from the outset.

The interpretation of null results becomes even more complicated in the setting of replication studies. In a replication study, researchers attempt to repeat an original study as closely as possible in order to assess whether consistent results can be obtained with new data (*National Academies of Sciences, Engineering, and Medicine, 2019*). In the last decade, various large-scale replication projects have been conducted in diverse fields, from the biomedical to the social sciences (*Prinz et al., 2011*; *Begley and Ellis, 2012*; *Klein et al., 2014*; *Open Science Collaboration, 2015*; *Camerer et al., 2016*; *Camerer et al., 2018*; *Klein et al., 2018*; *Cova et al., 2021*; *Errington et al., 2021*, among others). The majority of these projects reported alarmingly low replicability rates across a broad spectrum of criteria for quantifying replicability. While most of these projects restricted their focus on original studies with statistically significant results ('positive results'), the *Reproducibility Project: Cancer Biology* (RPCB, *Errington et al., 2021*), the *Experimental Philosophy Replicability Project* (EPRP, *Cova et al., 2021*), and the *Reproducibility Project: Psychology* (RPP, *Open Science Collaboration, 2015*) also attempted to replicate some original studies with null results – either non-significant or interpreted as showing no evidence for a meaningful effect by the original authors.

Although the EPRP and RPP interpreted non-significant results in both original and replication studies as a 'replication success' for some individual replications (see, for example, the replication of *McCann, 2005*, replication report: https://osf.io/wcm7n, or the replication of *Ranganath and Nosek, 2008*, replication report: https://osf.io/9xt25), they excluded the original null results in the calculation of an overall replicability rate based on significance. In contrast, the RPCB explicitly defined null results in both the original and the replication study as a criterion for 'replication success.' According to this 'non-significance' criterion, 11/15=73% replications of original null effects were successful. Four additional criteria were used to provide a more nuanced assessment of replication success for original null results: (i) whether the original effect estimate was included in the 95% confidence interval of the replication effect estimate (success rate 11/15=73%), (ii) whether the replication effect estimate was included in the 95% confidence interval of the original effect estimate (success rate 12/15=80%), (iii) whether the replication effect estimate was included in the 95% prediction interval based on the original effect estimate (success rate 12/15=80%), (iv) and whether the *p*-value obtained from combining the original and replication effect estimate with a meta-analysis was non-significant (success rate 10/15=67%). Criteria (i) to (iii) are useful for assessing compatibility in effect estimates between the original and the replication study. Their suitability has been extensively discussed in the literature. The prediction interval criterion (iii) or equivalent criteria (e.g. the *Q*-test) are usually recommended because they account for the uncertainty from both studies and have adequate error rates when the true effect sizes are the same (*Patil et al., 2016*; *Mathur and VanderWeele, 2020*; *Schauer and Hedges, 2021*).

While the effect estimate criteria (i) to (iii) can be applied regardless of whether or not the original study was non-significant, the 'meta-analytic non-significance' criterion (iv) and the aforementioned non-significance criterion refer specifically to original null results. We believe that there are several logical problems with both, and that it is important to highlight and address them, especially since the non-significance criterion has already been used in three replication projects without much scrutiny. It is crucial to note that it is not our intention to diminish the enormously important contributions of the RPCB, the EPRP, and the RPP, but rather to build on their work and provide recommendations for ongoing and future replication projects (e.g. *Amaral et al., 2019*; *Murphy et al., 2023*).

The logical problems with the non-significance criterion are as follows: First, if the original study had low statistical power, a non-significant result is highly inconclusive and does not provide evidence for the absence of an effect. It is then unclear what exactly the goal of the replication should be – to replicate the inconclusiveness of the original result? On the other hand, if the original study was adequately powered, a non-significant result may indeed provide some evidence for the absence of an effect when analyzed with appropriate methods, so that the goal of the replication is clearer. However, the criterion by itself does not distinguish between these two cases. Second, with this criterion researchers can virtually always achieve replication success by conducting a replication study with a very small sample size, such that the *p*-value is non-significant and the result is inconclusive. This is because the null hypothesis under which the *p*-value is computed is misaligned with the goal of inference, which is to quantify the evidence for the absence of an effect. Third, the criterion does not

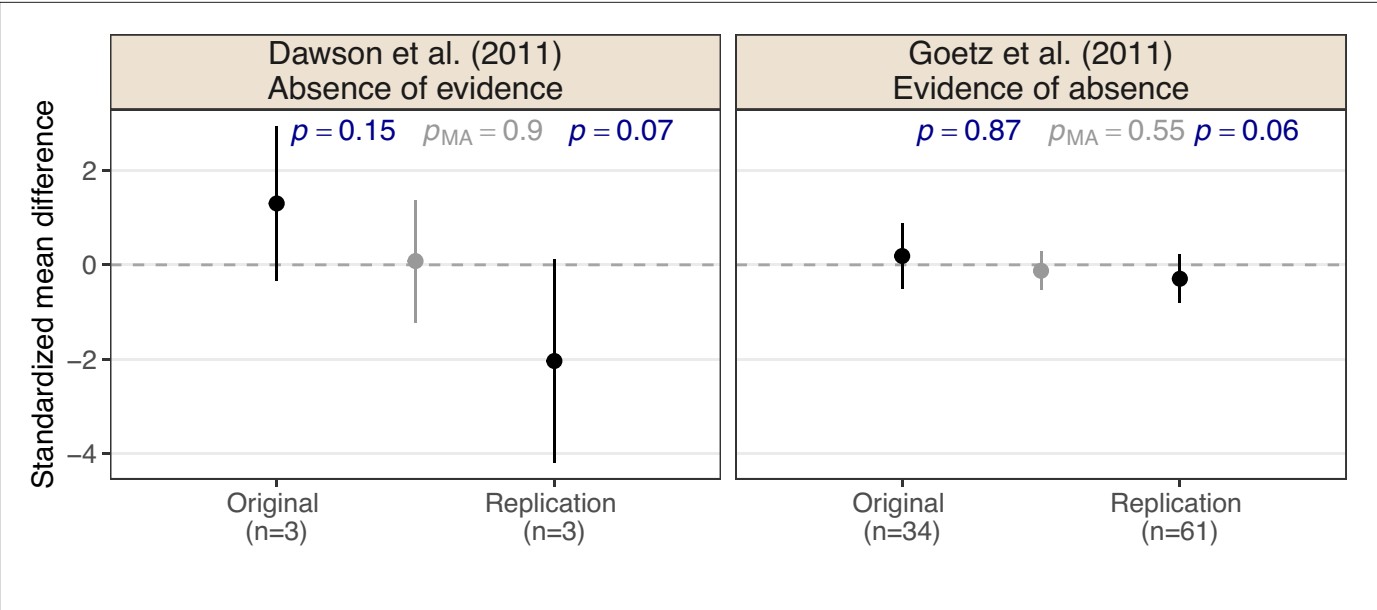

**Figure 1.** Two examples of original and replication study pairs which meet the non-significance replication success criterion from the Reproducibility Project: Cancer Biology (*Errington et al., 2021*). Shown are standardized mean difference effect estimates with 95% confidence intervals, sample sizes *n*, and two-sided *p*-values *p* for the null hypothesis that the effect is absent. Effect estimate, 95% confidence interval, and *p*-value from a fixed-effect meta-analysis $p_{MA}$ of original and replication study are shown in gray.

control the error of falsely claiming the absence of an effect at a predetermined rate. This is in contrast to the standard criterion for replication success, which requires significance from both studies (also known as the two-trials rule, see Section 12.2.8 in *Senn, 2021*), and ensures that the error of falsely claiming the presence of an effect is controlled at a rate equal to the squared significance level (for example, 5% × 5%=0.25% for a 5% significance level). The non-significance criterion may be intended to complement the two-trials rule for null results. However, it fails to do so in this respect, which may be required by regulators and funders. These logical problems are equally applicable to the meta-analytic non-significance criterion.

In the following, we present two principled approaches for analyzing replication studies of null results – frequentist equivalence testing and Bayesian hypothesis testing – that can address the limitations of the non-significance criterion. We use the null results replicated in the RPCB, RPP, and EPRP to illustrate the problems of the non-significance criterion and how they can be addressed. We conclude the paper with practical recommendations for analyzing replication studies of original null results, including simple R code for applying the proposed methods.

## Null results from the Reproducibility Project: Cancer Biology

*Figure 1* shows effect estimates on the standardized mean difference (SMD) scale with a 95% confidence intervals from two RPCB study pairs. In both study pairs, the original, and replication studies are 'null results' and, therefore, meet the non-significance criterion for replication success (the two-sided *p*-values are greater than 0.05 in both the original and the replication study). The same is true when applying the meta-analytic non-significance criterion (the two-sided *p*-values of the meta-analyses $p_{MA}$ are greater than 0.05). However, intuition would suggest that the conclusions in the two pairs are very different.

The original study from *Dawson et al., 2011* and its replication both show large effect estimates in magnitude, but due to the very small sample sizes, the uncertainty of these estimates is large, too. With such low sample sizes, the results seem inconclusive. In contrast, the effect estimates from *Goetz et al., 2011* and its replication are much smaller in magnitude and their uncertainty is also smaller because the studies used larger sample sizes. Intuitively, the results seem to provide more evidence for a zero (or negligibly small) effect. While these two examples show the qualitative difference

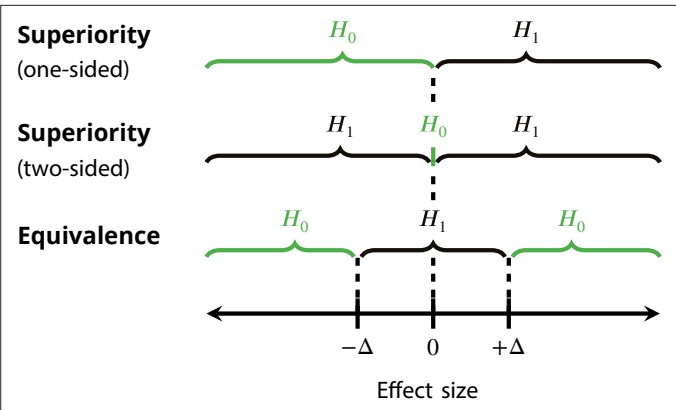

**Figure 2.** Null hypothesis ($H_0$) and alternative hypothesis ($H_1$) for superiority and equivalence tests (with equivalence margin $\Delta > 0$).

between the absence of evidence and evidence of absence, we will now discuss how the two can be quantitatively distinguished.

## Methods for assessing replicability of null results

There are both frequentist and Bayesian methods that can be used for assessing evidence for the absence of an effect. *Anderson and Maxwell, 2016* provide an excellent summary in the context of replication studies in psychology. We now briefly discuss two possible approaches – frequentist equivalence testing and Bayesian hypothesis testing – and their application to the RPCB, EPRP, and RPP data.

### Frequentist equivalence testing

Equivalence testing was developed in the context of clinical trials to assess whether a new treatment – typically cheaper or with fewer side effects than the established treatment – is practically equivalent to the established treatment (*Wellek, 2010*; *Lakens, 2017*). The method can also be used to assess whether an effect is practically equivalent to an absent effect, usually zero. Using equivalence testing as a way to put non-significant results into context has been suggested by several authors (*Hauck and Anderson, 1986*; *Campbell and Gustafson, 2018*). The main challenge is to specify the margin $\Delta > 0$ that defines an equivalence range $[-\Delta, +\Delta]$ in which an effect is considered as absent for practical purposes. The goal is then to reject the null hypothesis that the true effect is outside the equivalence range. This is in contrast to the usual null hypotheses of superiority tests which state that the effect is zero or smaller than zero, see *Figure 2* for an illustration.

To ensure that the null hypothesis is falsely rejected at most $\alpha \times 100\%$ of the time, the standard approach is to declare equivalence if the $(1 - 2\alpha) \times 100\%$ confidence interval for the effect is contained within the equivalence range, for example, a 90% confidence interval for $\alpha = 0.05$ (*Westlake, 1972*). This procedure is equivalent to declaring equivalence when two one-sided tests (TOST) for the null hypotheses of the effect being greater/smaller than $+\Delta$ and $-\Delta$, are both significant at level $\alpha$ (*Schuirmann, 1987*). A quantitative measure of evidence for the absence of an effect is then given by the maximum of the two one-sided *p*-values – the TOST *p*-value (*Greenland, 2023*, section 4.4). In case a dichotomous replication success criterion for null results is desired, it is natural to require that both the original and the replication TOST *p*-values are smaller than some level α (conventionally $\alpha = 0.05$). Equivalently, the criterion would require the $(1 - 2\alpha) \times 100\%$ confidence intervals of the original and the replication to be included in the equivalence region. In contrast to the non-significance criterion, this criterion controls the error of falsely claiming replication success at level $\alpha^2$ when there is a true effect outside the equivalence margin, thus complementing the usual two-trials rule in drug regulation (*Senn, 2021*, Section 12.2.8).

Returning to the RPCB data, *Figure 3* shows the standardized mean difference effect estimates with 90% confidence intervals for all 15 effects which were treated as null results by the RPCB. [There

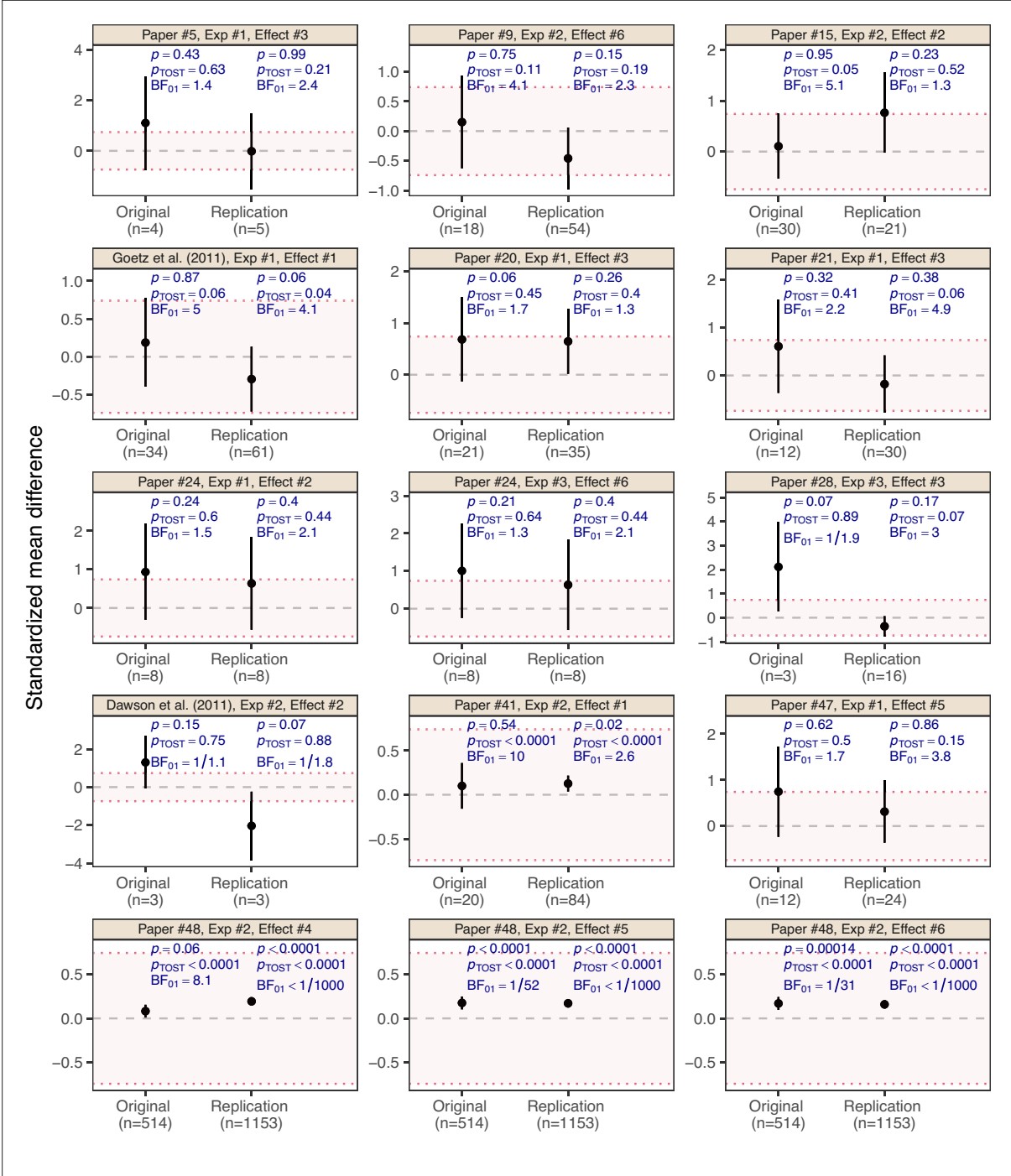

**Figure 3.** Effect estimates on the standardized mean difference (SMD) scale with 90% confidence interval for the 15 'null results' and their replication studies from the Reproducibility Project: Cancer Biology (*Errington et al., 2021*). The title above each plot indicates the original paper, experiment and effect numbers. Two original effect estimates from original paper 48 were statistically significant at $p < 0.05$, but were interpreted as null results by the original authors and, therefore, treated as null results by the RPCB. The two examples from *Figure 1* are indicated in the plot titles. The dashed gray line represents the value of no effect (SMD = 0), while the dotted red lines represent the equivalence range with a margin of $\Delta = 0.74$, classified as 'liberal' by *Wellek, 2010*, Table 1.1. The $p$-value $p_{TOST}$ is the maximum of the two one-sided $p$-values for the null hypotheses of the effect being greater/less than $+\Delta$ and $-\Delta$, respectively. The Bayes factor $BF_{01}$ quantifies the evidence for the null hypothesis $H_0: SMD = 0$ against the alternative $H_1: SMD \neq 0$ with normal unit-information prior assigned to the SMD under $H_1$.

are four original studies with null effects for which two or three 'internal' replication studies were conducted, leading in total to 20 replications of null effects. As done in the RPCB main analysis (**Errington et al., 2021**), we aggregated their SMD estimates into a single SMD estimate with fixed-effect meta-analysis and recomputed the replication *p*-value based on a normal approximation. For the original studies and the single replication studies we report the SMD estimates and *p*-values as provided by the RPCB]. Most of them showed non-significant *p*-values ($p > 0.05$) in the original study. It is noteworthy, however, that two effects from the second experiment of the original paper 48 were regarded as null results despite their statistical significance. According to the non-significance criterion (requiring $p > 0.05$ in original and replication study), there are 11 'successes' out of total 15 null effects, as reported in Table 1 from **Errington et al., 2021**.

We will now apply equivalence testing to the RPCB data. The dotted red lines in **Figure 3** represent an equivalence range for the margin $\Delta = 0.74$, which (**Wellek, 2010**, Table 1.1) classifies as 'liberal.' However, even with this generous margin, only 4 of the 15 study pairs are able to establish replication success at the 5% level, in the sense that both the original and the replication 90% confidence interval fall within the equivalence range (or, equivalently, that their TOST *p*-values are smaller than 0.05). For the remaining 11 studies, the situation remains inconclusive and there is no evidence for the absence or the presence of the effect. For instance, the previously discussed example from **Goetz et al., 2011** marginally fails the criterion ($p_{\text{TOST}} = 0.06$ in the original study and $p_{\text{TOST}} = 0.04$ in the replication), while the example from **Dawson et al., 2011** is a clearer failure ($p_{\text{TOST}} = 0.75$ in the original study and $p_{\text{TOST}} = 0.88$ in the replication) as both effect estimates even lie outside the equivalence margin.

The post-hoc specification of equivalence margins is controversial. Ideally, the margin should be specified on a case-by-case basis in a pre-registered protocol before the studies are conducted by researchers familiar with the subject matter. In the social and medical sciences, the conventions of **Cohen, 1992** are typically used to classify SMD effect sizes (SMD = 0.2 small, SMD = 0.5 medium, SMD = 0.8 large). While effect sizes are typically larger in preclinical research, it seems unrealistic to specify margins larger than 1 on the SMD scale to represent effect sizes that are absent for practical purposes. It could also be argued that the chosen margin $\Delta = 0.74$ is too lax compared to margins commonly used in clinical research (**Lange and Freitag, 2005**). We, therefore, report a sensitivity analysis regarding the choice of the margin in **Appendix 1—figure 1**. This analysis shows that for realistic margins between 0 and 1, the proportion of replication successes remains below 50% for the conventional $\alpha = 0.05$ level. To achieve a success rate of 11/15=73%, as was achieved with the non-significance criterion from the RPCB, unrealistic margins of $\Delta > 2$ are required.

Appendix 2 shows similar equivalence test analyses for the four study pairs with original null results from the RPP and EPRP. Three study pair results turn out to be inconclusive due to the large uncertainty around their effect estimates.

## Bayesian hypothesis testing

The distinction between absence of evidence and evidence of absence is naturally built into the Bayesian approach to hypothesis testing. A central measure of evidence is the Bayes factor (**Kass and Raftery, 1995**; **Goodman, 1999**; **Dienes, 2014**; **Keysers et al., 2020**), which is the updating factor of the prior odds to the posterior odds of the null hypothesis $H_0$ versus the alternative hypothesis $H_1$

$$\underbrace{\frac{\Pr(H_0 \text{ given data})}{\Pr(H_1 \text{ given data})}}_{\text{Posterior odds}} = \underbrace{\frac{\Pr(H_0)}{\Pr(H_1)}}_{\text{Prior odds}} \times \underbrace{\frac{\Pr(\text{data given } H_0)}{\Pr(\text{data given } H_1)}}_{\text{Bayes factor BF}_{01}}.$$

The Bayes factor $\text{BF}_{01}$ quantifies how much the observed data have increased or decreased the probability $\Pr(H_0)$ of the null hypothesis relative to the probability $\Pr(H_1)$ of the alternative. As such, Bayes factors are direct measures of evidence for the null hypothesis, in contrast to *p*-values, which are only indirect measures of evidence as they are computed under the assumption that the null hypothesis is true (**Held and Ott, 2018**). If the null hypothesis states the absence of an effect, a Bayes factor greater than one ($\text{BF}_{01} > 1$) indicates evidence for the absence of the effect and a Bayes factor smaller than one indicates evidence for the presence of the effect ($\text{BF}_{01} < 1$), whereas a Bayes factor not much different from one indicates absence of evidence for either hypothesis ($\text{BF}_{01} \approx 1$). Bayes factors are quantitative summaries of the evidence provided by the data in favor of the null hypothesis as opposed to the alternative hypothesis. However, if a dichotomous criterion for successful replication

of a null result is desired, it seems natural to require a Bayes factor larger than some level $\gamma > 1$ from both studies, for example, $\gamma = 3$ or $\gamma = 10$ which are conventional levels for 'substantial' and 'strong' evidence, respectively (*Jeffreys, 1961*). In contrast to the non-significance criterion, this criterion provides a genuine measure of evidence that can distinguish absence of evidence from evidence of absence.

The main challenge with Bayes factors is the specification of the effect under the alternative hypothesis $H_1$. The assumed effect under $H_1$ is directly related to the Bayes factor, and researchers who assume different effects will end up with different Bayes factors. Instead of specifying a single effect, one therefore typically specifies a 'prior distribution' of plausible effects. Importantly, the prior distribution, like the equivalence margin, should be determined by researchers with subject knowledge and before the data are collected.

To compute the Bayes factors for the RPCB null results, we used the observed effect estimates as the data and assumed a normal sampling distribution for them (*Dienes, 2014*), as typically done in a meta-analysis. The Bayes factors $\mathrm{BF}_{01}$ shown in *Figure 3* then quantify the evidence for the null hypothesis of no effect against the alternative hypothesis that there is an effect using a normal 'unit-information' prior distribution (*Kass and Wasserman, 1995*) for the effect size under the alternative $H_1$, see Appendix 3 for further details on the calculation of these Bayes factors. We see that in most cases there is no substantial evidence for either the absence or the presence of an effect, as with the equivalence tests. For instance, with a lenient Bayes factor threshold of 3, only 1 of the 15 replications are successful, in the sense of having $\mathrm{BF}_{01} > 3$ in both the original and the replication study. The Bayes factors for the two previously discussed examples are consistent with our intuitions – in the *Goetz et al., 2011* example there is indeed substantial evidence for the absence of an effect ($\mathrm{BF}_{01} = 5$ in the original study and $\mathrm{BF}_{01} = 4.1$ in the replication), while in the *Dawson et al., 2011* example there is even anecdotal evidence for the *presence* of an effect, though the Bayes factors are very close to one due to the small sample sizes ($\mathrm{BF}_{01} = 1/1.1$ in the original study and $\mathrm{BF}_{01} = 1/1.8$ in the replication).

As with the equivalence margin, the choice of the prior distribution for the SMD under the alternative $H_1$ is debatable. The normal unit-information prior seems to be a reasonable default choice, as it implies that small to large effects are plausible under the alternative, but other normal priors with smaller/larger standard deviations could have been considered to make the test more sensitive to smaller/larger true effect sizes. The sensitivity analysis in Appendix 1, therefore, also includes an analysis on the effect of varying prior standard deviations and the Bayes factor thresholds. However, again, to achieve replication success for a larger proportion of replications than the observed 1/15=7%, unreasonably large prior standard deviations have to be specified.

Of note, among the 15 RPCB null results, there are three interesting cases (the three effects from original paper 48 by *Lin et al., 2012* and its replication by *Lewis et al., 2018*) where the Bayes factor is qualitatively different from the equivalence test, revealing a fundamental difference between the two approaches. The Bayes factor is concerned with testing whether the effect is *exactly zero*, whereas the equivalence test is concerned with whether the effect is within an *interval around zero*. Due to the very large sample size in the original study (n=514) and the replication (n=1153), the data are incompatible with an exactly zero effect, but compatible with effects within the equivalence range. Apart from this example, however, both approaches lead to the same qualitative conclusion – most RPCB null results are highly ambiguous.

Appendix 2 also shows Bayes factor analyses for the four study pairs with original null results from the RPP and EPRP. In contrast to the RPCB results, most Bayes factors indicate non-anecdotal evidence for a null effect in cases where the non-significance criterion was met, possibly because of the larger sample sizes and smaller effects in these fields.

## Conclusions

The concept of 'replication success' is inherently multifaceted. Reducing it to a single criterion seems to be an oversimplification. Nevertheless, we believe that the 'non-significance' criterion – declaring a replication as successful if both the original and the replication study produce non-significant results – is not fit for purpose. This criterion does not ensure that both studies provide evidence for the absence of an effect, it can be easily achieved for any outcome if the studies have sufficiently small sample sizes, and it does not control the relevant error rates. While it is important to replicate original

studies with null results, we believe that they should be analyzed using more informative approaches. *Box 1* summarizes our recommendations.

Our reanalysis of the RPCB studies with original null results showed that for most studies that meet the non-significance criterion, the conclusions are much more ambiguous – both with frequentist and Bayesian analyses. While the exact success rate depends on the equivalence margin and the prior distribution, our sensitivity analyses show that even with unrealistically liberal choices, the success rate remains below 40% which is substantially lower than the 73% success rate based on the non-significance criterion.

This is not unexpected, as a study typically requires larger sample sizes to detect the absence of an effect than to detect its presence (*Matthews, 2006*, Section 11.5.3). Of note, the RPCB sample sizes were chosen so that each replication had at least 80% power to detect the original effect estimate based on a standard superiority test. However, the design of replication studies should ideally align with the planned analysis (*Anderson and Kelley, 2022*) so if the goal of the study is to find evidence for the absence of an effect, the replication sample size should be determined based on a test for equivalence, see *Flight and Julious, 2016* and *Pawel et al., 2023a* for frequentist and Bayesian approaches, respectively.

Our reanalysis of the RPP and EPRP studies with original null results showed that Bayes factors indeed indicate some evidence for no effect in cases where the non-significance criterion was satisfied, possibly due to the smaller effects and typically larger sample sizes in these fields compared to cancer biology. On the other hand, in most cases, the precision of the effect estimates was still limited so that only one study pair achieved replication success with the equivalence testing approach. However, it is important to note that the conclusions from the RPP and EPRP analyses are merely anecdotal, as there were only four study pairs with original null results to analyze.

For both the equivalence test and the Bayes factor approach, it is critical that the equivalence margin and the prior distribution are specified independently of the data, ideally before the original and replication studies are conducted. Typically, however, the original studies were designed to find evidence for the presence of an effect, and the goal of replicating the 'null result' was formulated only after failure to do so. It is, therefore, important that margins and prior distributions are motivated from historical data and/or field conventions (*Campbell and Gustafson, 2021*), and that sensitivity analyses regarding their choice are reported.

In addition, when analyzing a single pair of original and replication studies, we recommend interpreting Bayes factors and TOST $p$-values as quantitative measures of evidence and discourage dichotomizing them into 'success' or 'failure.' For example, two TOST $p$-values $p_{\text{TOST}} = 0.049$ and $p_{\text{TOST}} = 0.051$ carry similar evidential weight regardless of one being slightly smaller and the other being slightly larger than 0.05. On the other hand, when more than one pair of original and replication studies are analyzed, dichotomization may be required for computing an overall success rate. In this case, the rate may be computed for different thresholds that correspond to qualitatively different levels of evidence (e.g. 1, 3, and 10 for Bayes factors, or 0.05, 0.01, and 0.005 for $p$-values).

Researchers may also ask whether the equivalence test or the Bayes factor is 'better.' We believe that this is the wrong question to ask, because both methods address different questions and are better in different senses; the equivalence test is calibrated to have certain frequentist error rates, which the Bayes factor is not. The Bayes factor, on the other hand, seems to be a more natural measure of evidence as it treats the null and alternative hypotheses symmetrically and represents the factor by which rational agents should update their beliefs in light of the data. Replication success is ideally evaluated along multiple dimensions, as nicely exemplified by the RPCB, EPRP, and RPP. Replications that are successful on multiple criteria provide more convincing support for the original finding, while replications that are successful on fewer criteria require closer examination. Fortunately, the use of multiple methods is already standard practice in replication assessment, so our proposal to use both of them does not require a major paradigm shift.

While the equivalence test and the Bayes factor are two principled methods for analyzing original and replication studies with null results, they are not the only possible methods for doing so. A straightforward extension would be to first synthesize the original and replication effect estimates with a meta-analysis, and then apply the equivalence and Bayes factor tests to the meta-analytic estimate similar to the meta-analytic non-significance criterion used by the RPCB. This could potentially improve the power of the tests, but consideration must be given to the threshold used for the

## Box 1. Recommendations for the analysis of replication studies of original null results.

Calculations are based on effect estimates $\hat{\theta}_i$ with standard errors $\sigma_i$ from an original study ($i = o$) and its replication ($i = r$). Both effect estimates are assumed to be normally distributed around the true effect size $\theta$ with known variance $\sigma_i^2$. The effect size $\theta_0$ represents the value of no effect, typically $\theta_0 = 0$.

**Equivalence test**

1. Specify a margin $\Delta > 0$ that defines an equivalence range $[\theta_0 - \Delta, \theta_0 + \Delta]$ in which effects are considered absent for practical purposes.

2. Compute the TOST $p$-values for original ($i = o$) and replication ($i = r$) data

$$p_{\text{TOST},i} = \max \left\{ \Phi \left( \frac{\hat{\theta}_i - \theta_0 - \Delta}{\sigma_i} \right), 1 - \Phi \left( \frac{\hat{\theta}_i - \theta_0 + \Delta}{\sigma_i} \right) \right\},$$

with $\Phi(\cdot)$ the cumulative distribution function of the standard normal distribution.

```
## R function to compute TOST p-value based on effect estimate, standard
## error, null value (default is 0), equivalence margin from step 1.
pTOST <- function(estimate, se, null = 0, margin) {
 p1 <- pnorm(q = (estimate - null - margin)/se)
 p2 <- 1 - pnorm(q = (estimate - null + margin)/se)
 p <- pmax(p1, p2)
 return(p)
}
```

3. Declare replication success at level $\alpha$ if $p_{\text{TOST},o} \leq \alpha$ and $p_{\text{TOST},r} \leq \alpha$, conventionally $\alpha = 0.05$.

4. Perform a sensitivity analysis with respect to the margin $\Delta$. For example, visualize the TOST $p$-values for different margins to assess the robustness of the conclusions.

**Bayes factor**

1. Specify a prior distribution for the effect size $\theta$ that represents plausible values under the alternative hypothesis that there is an effect ($H_1 : \theta \neq \theta_0$). For example, specify the mean $m$ and standard deviation $s$ of a normal distribution $\theta | H_1 \sim \text{N}(m, s^2)$.

2. Compute the Bayes factors contrasting $H_0 : \theta = \theta_0$ to $H_1 : \theta \neq \theta_0$ for original ($i = o$) and replication ($i = r$) data. Assuming a normal prior distribution, the Bayes factor is

$$\text{BF}_{01,i} = \sqrt{1 + \frac{s^2}{\sigma_i^2}} \exp \left[ -\frac{1}{2} \left\{ \frac{(\hat{\theta}_i - \theta_0)^2}{\sigma_i^2} - \frac{(\hat{\theta}_i - m)^2}{\sigma_i^2 + s^2} \right\} \right].$$

```
## R function to compute Bayes factor based on effect estimate, standard
## error, null value (default is 0), prior mean (default is null value),
## and prior standard deviation from step 1.
BF01 <- function(estimate, se, null = 0, priormean = null, priorsd) {
 bf <- sqrt(1 + priorsd^2/se^2)*exp(-0.5*((estimate - null)^2/se^2 -
 (estimate - priormean)^2/(se^2 + priorsd^2)))
```

```
  return(bf)
}
```

3. Declare replication success at level $\gamma > 1$ if $\mathrm{BF}_{01,o} \geq \gamma$ and $\mathrm{BF}_{01,r} \geq \gamma$, conventionally $\gamma = 3$ (substantial evidence) or $\gamma = 10$ (strong evidence).

4. Perform a sensitivity analysis with respect to the prior distribution. For example, visualize the Bayes factors for different prior standard deviations to assess the robustness of the conclusions.

$p$-values/Bayes factors, as naive use of the same thresholds as in the standard approaches may make the tests too liberal (*Shun et al., 2005*). Furthermore, there are various advanced methods for quantifying evidence for absent effects which could potentially improve on the more basic approaches considered here (*Lindley, 1998*; *Johnson and Rossell, 2010*; *Morey and Rouder, 2011*; *Kruschke, 2018*; *Stahel, 2021*; *Micheloud and Held, 2023*; *Izbicki et al., 2023*).

## Acknowledgements

We thank the RPCB, EPRP, and RPP contributors for their tremendous efforts and for making their data publicly available. We thank Maya Mathur for her helpful advice on data preparation. We thank Benjamin Ineichen for helpful comments on drafts of the manuscript. We thank the three reviewers and the reviewing editor for useful comments that substantially improved the paper. Our acknowledgment of these individuals does not imply their endorsement of our work. We thank the Swiss National Science Foundation for financial support (grant #189295).

## Additional information

### Funding

| Funder | Grant reference number | Author |
| --- | --- | --- |
| Schweizerischer Nationalfonds zur Förderung der Wissenschaftlichen Forschung | 189295 | Leonhard Held |

The funders had no role in study design, data collection and interpretation, or the decision to submit the work for publication.

### Author contributions

Samuel Pawel, Rachel Heyard, Conceptualization, Resources, Data curation, Software, Formal analysis, Validation, Investigation, Visualization, Methodology, Writing – original draft, Project administration, Writing – review and editing; Charlotte Micheloud, Conceptualization, Resources, Data curation, Formal analysis, Validation, Investigation, Methodology, Writing – review and editing; Leonhard Held, Conceptualization, Resources, Supervision, Funding acquisition, Validation, Investigation, Methodology, Project administration, Writing – review and editing

### Author ORCIDs

Samuel Pawel ⬥ https://orcid.org/0000-0003-2779-320X
Rachel Heyard ⬥ http://orcid.org/0000-0002-7531-4333
Charlotte Micheloud ⬥ http://orcid.org/0000-0002-4995-4505
Leonhard Held ⬥ http://orcid.org/0000-0002-8686-5325

Reviewer #1 (Public review): https://doi.org/10.7554/eLife.92311.3.sa1
Reviewer #2 (Public review): https://doi.org/10.7554/eLife.92311.3.sa2

Author response https://doi.org/10.7554/eLife.92311.3.sa3

## Additional files

### Supplementary files
• MDAR checklist

### Data availability
The code and data to reproduce our analyses is openly available at https://gitlab.uzh.ch/samuel.pawel/rsAbsence. A snapshot of the repository at the time of writing is archived at https://doi.org/10.5281/zenodo.7906792 (*Pawel et al., 2023b*). We used the statistical programming language R version 4.3.2 (*R Development Core Team, 2022*) for analyses. The R packages ggplot2 (*Wickham, 2016*), dplyr (*Wickham et al., 2022*), knitr (*Xie, 2022*), and reporttools (*Rufibach, 2009*) were used for plotting, data preparation, dynamic reporting, and formatting, respectively. The data from the RPCB were obtained by downloading the files from https://github.com/mayamathur/rpcb (*Mathur, 2022*; commit a1e0c63) and extracting the relevant variables as indicated in the R script preprocess-rpcb-data.R which is available in our git repository. The RPP and EPRP data were obtained from the RProjects data set available in the R package ReplicationSuccess (*Held, 2020*), see the package documentation (https://CRAN.R-project.org/package=ReplicationSuccess) for details on data extraction.

The following previously published datasets were used:

| Author(s) | Year | Dataset title | Dataset URL | Database and Identifier |
|---|---|---|---|---|
| Pawel S, Held L | 2020 | Data from four large-scale replication projects | https://doi.org/10.5281/zenodo.10350448 | Zenodo, 10.5281/zenodo.10350448 |
| Errington TM, Mathur M, Soderberg CK, Denis A, Perfito N, Iorns E, Nosek BA | 2021 | Data from the Reproducibility Project: Cancer Biology | https://doi.org/10.17605/OSF.IO/SQUY7 | Open Science Framework, 10.17605/OSF.IO/SQUY7 |

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

# Appendix 1

## Sensitivity analyses

The post-hoc specification of equivalence margins $\Delta$ and prior distribution for the SMD under the alternative $H_1$ is debatable. Commonly used margins in clinical research are much more stringent (**Lange and Freitag, 2005**); for instance, in oncology, a margin of $\Delta = \log(1.3)$ is commonly used for log odds/hazard ratios, whereas in bioequivalence studies a margin of $\Delta = \log(1.25)$ is the convention (**Senn, 2021**, Chapter 22). These margins would translate into margins of $\Delta = 0.14$ and $\Delta = 0.12$ on the SMD scale, respectively, using the $\text{SMD} = (\sqrt{3}/\pi) \log \text{OR}$ conversion (**Cooper et al., 2019**, p. 233). Similarly, for the Bayes factor we specified a normal unit-information prior under the alternative while other normal priors with smaller/larger standard deviations could have been considered. Here, we therefore investigate the sensitivity of our conclusions with respect to these parameters.

The top plot of **Appendix 1—figure 1** shows the number of successful replications as a function of the margin $\Delta$ and for different TOST $p$-value thresholds. Such an 'equivalence curve' approach was first proposed by **Hauck and Anderson, 1986**. We see that for realistic margins between 0 and 1, the proportion of replication successes remains below 50% for the conventional $\alpha = 0.05$ level. To achieve a success rate of 11/15=73%, as was achieved with the non-significance criterion from the RPCB, unrealistic margins of $\Delta > 2$ are required. Changing the success criterion to a more lenient level ($\alpha = 0.1$) or a more stringent level ($\alpha = 0.01$) hardly changes the conclusion.

The bottom plot of **Appendix 1—figure 1** shows a sensitivity analysis regarding the choice of the prior standard deviation and the Bayes factor threshold. It is uncommon to specify prior standard deviations larger than the unit-information standard deviation of 2, as this corresponds to the assumption of very large effect sizes under the alternative. However, to achieve replication success for a larger proportion of replications than the observed 1/15=7%, unreasonably large prior standard deviations have to be specified. For instance, a standard deviation of roughly 5 is required to achieve replication success in 50% of the replications at a lenient Bayes factor threshold of $\gamma = 3$. The standard deviation needs to be almost 20 so that the same success rate 11/15=73% as with the non-significance criterion is achieved. The necessary standard deviations are even higher for stricter Bayes factor thresholds, such as $\gamma = 6$ or $\gamma = 10$.

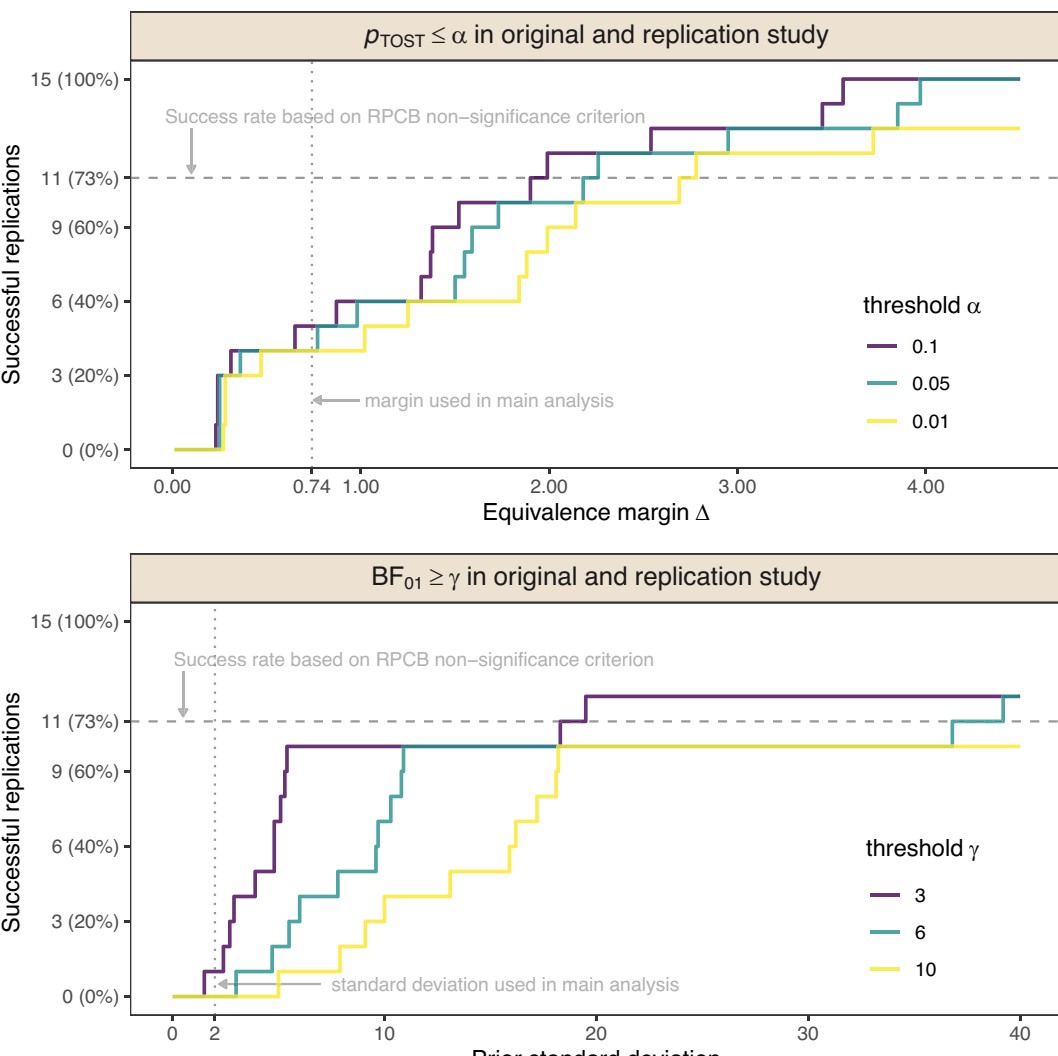

**Appendix 1—figure 1.** Number of successful replications of original null results in the Reproducibility Project: Cancer Biology (RPCB) as a function of the margin $\Delta$ of the equivalence test ($p_{\mathrm{TOST}} \leq \alpha$ in both studies for $\alpha = 0.1, 0.05, 0.01$) or the standard deviation of the zero-mean normal prior distribution for the standardized mean difference (SMD) effect size under the alternative $H_1$ of the Bayes factor test ($BF_{01} \geq \gamma$ in both studies for $\gamma = 3, 6, 10$).

## Appendix 2

### Null results from the RPP and EPRP

Here, we perform equivalence test and Bayes factor analyses for the three original null results from the Reproducibility Project: Psychology (*Eastwick and Finkel, 2008*; *Ranganath and Nosek, 2008*; *Reynolds and Besner, 2008*) and the original null result from the Reproducibility Project: Experimental Philosophy (*McCann, 2005*). To enable comparison of effect sizes across different studies, both the RPP and the EPRP provided effect estimates as Fisher *z*-transformed correlations which we use in the following.

*Appendix 2—figure 1* shows effect estimates with 90% confidence intervals, two-sided *p*-values for the null hypothesis that the effect size is zero, TOST *p*-values for a margin of $\Delta = 0.2$ , and Bayes factors using a normal prior centered around zero with a standard deviation of 2. We see that all replications except the replication of *Ranganath and Nosek, 2008* would be considered successful with the non-significance criterion, as the original and replication *p*-values are greater than 0.05. In all three cases, the Bayes factors also indicate substantial ($\mathrm{BF}_{01} > 3$) to strong evidence ($\mathrm{BF}_{01} > 10$) for the null hypothesis of no effect. Compared to the Bayes factors in the RPCB, the evidence is stronger, possibly due to the mostly larger sample sizes in the RPP and EPRP.

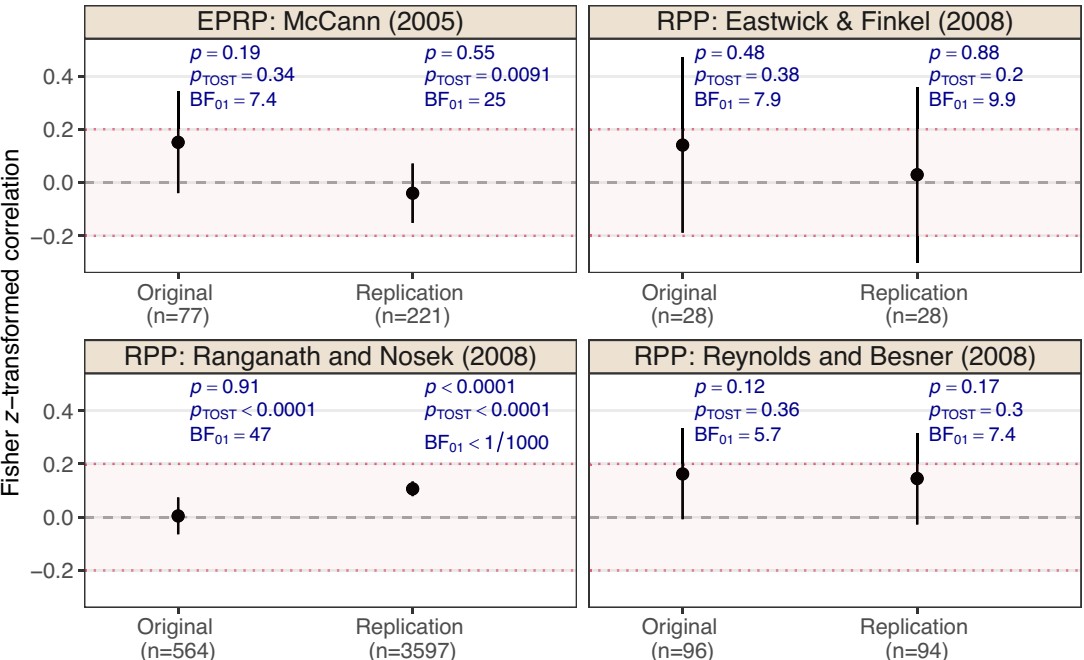

**Appendix 2—figure 1.** Effect estimates on Fisher *z*-transformed correlation scale with a 90% confidence interval for the 'null results' and their replication studies from the Reproducibility Project: Psychology (RPP, *Open Science Collaboration, 2015*) and the Experimental Philosophy Replicability Project (EPRP, *Cova et al., 2021*). The dashed gray line represents the value of no effect ($z = 0$), while the dotted red lines represent the equivalence range with a margin of $\Delta = 0.74$. The *p*-value $p_{\mathrm{TOST}}$ is the maximum of the two one-sided *p*-values for the null hypotheses of the effect being greater/less than $+\Delta$ and $-\Delta$, respectively. The Bayes factor $\mathrm{BF}_{01}$ quantifies the evidence for the null hypothesis $H_0 : z = 0$ against the alternative $H_1 : z \neq 0$ with normal prior centered around zero and standard deviation of 2 assigned to the effect size under $H_1$.

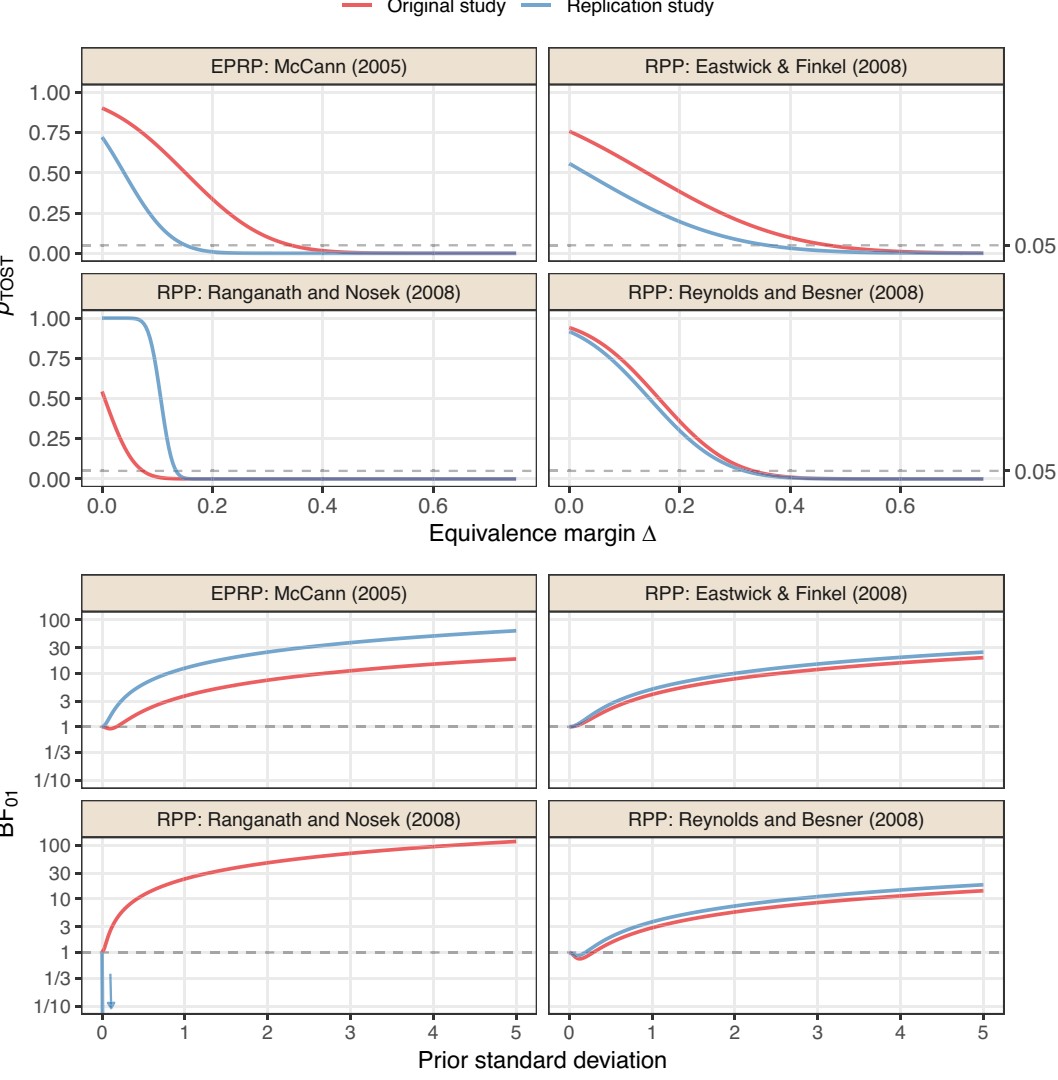

**Appendix 2—figure 2.** Sensitivity analyses for the 'null results' and their replication studies from the Reproducibility Project: Psychology (RPP, *Open Science Collaboration, 2015*) and the Experimental Philosophy Replicability Project (EPRP, *Cova et al., 2021*). The Bayes factor of the replication of *Ranganath and Nosek, 2008* decreases very quickly and is only shown for a limited range.

Interestingly, the opposite conclusion is reached when we analyze the data using an equivalence test with a margin of $\Delta = 0.2$ (which may be considered liberal as it represents a small to medium effect based on the *Cohen, 1992* convention). In this case, equivalence at the 5% level can only be established for the *Ranganath and Nosek, 2008* original study and its replication simultaneously, as the confidence intervals from the other studies are too wide to be included in the equivalence range. Furthermore, the Ranganath and Nosek (2008) replication also illustrates the conceptual difference between testing for an *exactly zero* effect versus testing for an effect *within an interval around zero*. That is, the Bayes factor indicates no evidence for a zero effect (because the estimate is clearly not zero), but the equivalence test indicates evidence for a negligible effect (because the estimate is clearly within the equivalence range).

As before, the particular choices of the equivalence margin $\Delta$ for the equivalence test and prior standard deviation of the Bayes factor are debatable. We, therefore, report sensitivity analyses in *Appendix 2—figure 2* which show the TOST *p*-values and Bayes factors of original and replication studies for a range of margins and prior standard deviations, respectively. Apart from the *Ranganath and Nosek, 2008* study pair, all studies require large margins of about $\Delta = 0.4$ to establish replication

success at the 5% level (in the sense of original and replication TOST $p$-values being smaller than 0.05). On the other hand, in all but the *Ranganath and Nosek, 2008* replication, the data provide substantial evidence for a null effect ($\mathrm{BF}_{01} > 3$) for prior standard deviations of about one, while larger prior standard deviations of about three are required for the data to indicate strong evidence ($\mathrm{BF}_{01} > 10$) for a null effect, whereas the data from the *Ranganath and Nosek, 2008* replication provide very strong evidence against a null effect for all prior standard deviations considered.

## Appendix 3

### Technical details on Bayes factors

We assume that effect estimates are normally distributed around an unknown effect size $\theta$ with known variance equal to their squared standard error, i.e.,

$$\hat{\theta}_i \mid \theta \sim \mathrm{N}(\theta, \sigma_i^2)$$

for original ($i = o$) and replication ($i = r$). This framework is similar to meta-analysis and can be applied to many types of effect sizes and data (*Spiegelhalter et al., 2003*, Section 2.4). We want to quantify the evidence for the null hypothesis that the effect size is equal to a null effect ($H_0\colon \theta = \theta_0$, typically $\theta_0 = 0$) against the alternative hypothesis that the effect size is non-null ($H_1\colon \theta \neq \theta_0$). This requires specification of a prior distribution for the effect size under the alternative, and we will assume a normal prior $\theta \mid H_1 \sim \mathrm{N}(m, s^2)$ in the following. The Bayes factor based on an effect estimate is then given by the ratio of its likelihood under the null hypothesis to its marginal likelihood under the alternative hypothesis, i.e.,

$$
\begin{aligned}
\mathrm{BF}_{01,i} &= \frac{p(\hat{\theta}_i \mid H_0)}{p(\hat{\theta}_i \mid H_1)} \\
&= \frac{p(\hat{\theta}_i \mid \theta_0)}{\int_{-\infty}^{+\infty} p(\hat{\theta}_i \mid \theta)\, p(\theta \mid H_1)\, \mathrm{d}\theta} \\
&= \sqrt{1 + \frac{s^2}{\sigma_i^2}}\, \exp\left[-\frac{1}{2}\left\{\frac{(\hat{\theta}_i - \theta_0)^2}{\sigma_i^2} - \frac{(\hat{\theta}_i - m)^2}{\sigma_i^2 + s^2}\right\}\right].
\end{aligned}
$$

In the main analysis, we used a normal unit-information prior, that is, a normal distribution centered around the value of no effect ($m = 0$) with a standard deviations corresponding to the standard error of an SMD estimate based on one observation (*Kass and Wasserman, 1995*). Assuming that the group means are normally distributed $\overline{X}_1 \sim \mathrm{N}(\theta_1, 2\tau^2/n)$ and $\overline{X}_2 \sim \mathrm{N}(\theta_2, 2\tau^2/n)$ with $n$ the total sample size and $\tau$ the known data standard deviation, the distribution of the SMD is $\mathrm{SMD} = (\overline{X}_1 - \overline{X}_2)/\tau \sim \mathrm{N}\{(\theta_1 - \theta_2)/\tau, \sigma^2 = 4/n\}$. The standard error $\sigma$ of the SMD based on one unit ($n = 1$), is hence 2, meaning that the standard deviation of the unit-information prior is $s = 2$.

