## [Editor Report · eLife assessment]

This work provides a **valuable** contribution and assessment of what it means to replicate a null study finding, and what are the appropriate methods for doing so (apart from a rote p-value assessment). Through a **convincing** re-analysis of results from the Reproducibility Project: Cancer Biology using frequentist equivalence testing and Bayes factors, the authors demonstrate that even when reducing 'replicability success' to a single criterion, how precisely replication is measured may yield differing results. Less focus is directed to appropriate replication of non-null findings.

---

## [Referee Report · Reviewer #1 (Public review)]

Summary:

The goal of Pawel et al. is to provide a more rigorous and quantitative approach for judging whether or not an initial null finding (conventionally with p >= 0.05) has been replicated by a second similarly null finding. They discuss important objections to relying on the qualitative significant/non-significant dichotomy to make this judgement. They present two complementary methods (one frequentist and the other Bayesian) which provide a superior quantitative framework for assessing the replicability of null findings.

Strengths:

Clear presentation; illuminating examples drawn from the well-known Reproducibility Project: Cancer Biology data set; R-code that implements suggested analyses. Using both methods as suggested provides a superior procedure for judging the replicability of null findings.

Weaknesses:

The frequentist and the Bayesian methods can be used to make binary assessments of an original finding and its replication. The authors clarify, though, that they can also be used to make continuous quantitative judgements about strength of evidence. I believe that most will use the methods in a binary fashion, but the availability of more nuanced assessments is welcome. This revision has addressed what I initially considered a weakness.

---

## [Referee Report · Reviewer #2 (Public review)]

Summary and strengths:

The work provides significant insights because usually non-significant studies can be considered replicated by their null replications as well. The work discuss and provide data demonstrating that when analyzing studies with p > 0.05 for the result to be replicated, equivalence tests and bayes factor approaches are more suitable, since studies can be underpowered even if replications use larger samples than their original studies in general. Non-significant p-values are highly expected even with 80% of power for a true effect.The evidence used features methods and analyses more rigorous than current state-of-the-art research on replicability.

I am satisfied with the revisions made by the authors in response to my initial suggestions, as well as their subsequent responses to my observations throughout the reviewing process.

---

## [Author Response]

The following is the authors’ response to the original reviews.

**eLife assessment**
This work provides a valuable contribution and assessment of what it means to replicate a null study finding, and what are the appropriate methods for doing so (apart from a rote p-value assessment). Through a convincing re-analysis of results from the Reproducibility Project: Cancer Biology using frequentist equivalence testing and Bayes factors, the authors demonstrate that even when reducing 'replicability success' to a single criterion, how precisely replication is measured may yield differing results. Less focus is directed to appropriate replication of non-null findings.
**Reviewer #1 (Public Review):**
Summary:The goal of Pawel et al. is to provide a more rigorous and quantitative approach for judging whether or not an initial null finding (conventionally with p ≥ 0.05) has been replicated by a second similarly null finding. They discuss important objections to relying on the qualitative significant/non-significant dichotomy to make this judgment. They present two complementary methods (one frequentist and the other Bayesian) which provide a superior quantitative framework for assessing the replicability of null findings.Strengths:Clear presentation; illuminating examples drawn from the well-known Reproducibility Project: Cancer Biology data set; R-code that implements suggested analyses. Using both methods as suggested provides a superior procedure for judging the replicability of null findings.Weaknesses:The proposed frequentist and the Bayesian methods both rely on binary assessments of an original finding and its replication. I'm not sure if this is a weakness or is inherent to making binary decisions based on continuous data.For the frequentist method, a null finding is considered replicated if the original and replication 90% confidence intervals for the effects both fall within the equivalence range. According to this approach, a null finding would be considered replicated if p-values of both equivalences tests (original and replication) were, say, 0.049, whereas would not be considered replicated if, for example, the equivalence test of the original study had a p-value of 0.051 and the replication had a p-value of 0.001. Intuitively, the evidence for replication would seem to be stronger in the second instance. The recommended Bayesian approach similarly relies on a dichotomy (e.g., Bayes factor > 1).

Thanks for the suggestions, we now emphasize more strongly in the “Methods for assessing replicability of null results” and “Conclusions” sections that both TOST p-values and Bayes factors are quantitative measures of evidence that do not require dichotomization into “success” or “failure”.

**Reviewer #2 (Public Review):**
Summary:The study demonstrates how inconclusive replications of studies initially with p > 0.05 can be and employs equivalence tests and Bayesian factor approaches to illustrate this concept. Interestingly, the study reveals that achieving a success rate of 11 out of 15, or 73%, as was accomplished with the non-significance criterion from the RPCB (Reproducibility Project: Cancer Biology), requires unrealistic margins of Δ > 2 for equivalence testing.Strengths:The study uses reliable and shareable/open data to demonstrate its findings, sharing as well the code for statistical analysis. The study provides sensitivity analysis for different scenarios of equivalence margin and alfa level, as well as for different scenarios of standard deviations for the prior of Bayes factors and different thresholds to consider. All analysis and code of the work is open and can be replicated. As well, the study demonstrates on a case-by-case basis how the different criteria can diverge, regarding one sample of a field of science: preclinical cancer biology. It also explains clearly what Bayes factors and equivalence tests are.Weaknesses:It would be interesting to investigate whether using Bayes factors and equivalence tests in addition to p-values results in a clearer scenario when applied to replication data from other fields. As mentioned by the authors, the Reproducibility Project: Experimental Philosophy (RPEP) and the Reproducibility Project: Psychology (RPP) have data attempting to replicate some original studies with null results. While the RPCB analysis yielded a similar picture when using both criteria, it is worth exploring whether this holds true for RPP and RPEP. Considerations for further research in this direction are suggested. Even if the original null results were excluded in the calculation of an overall replicability rate based on significance, sensitivity analyses considering them could have been conducted. The present authors can demonstrate replication success using the significance criteria in these two projects with initially p < 0.05 studies, both positive and non-positive.Other comments:Introduction: The study demonstrates how inconclusive replications of studies initially with p > 0.05 can be and employs equivalence tests and Bayesian factor approaches to illustrate this concept. Interestingly, the study reveals that achieving a success rate of 11 out of 15, or 73%, as was accomplished with the non-significance criterion from the RPCB (Reproducibility Project: Cancer Biology), requires unrealistic margins of Δ > 2 for equivalence testing.Overall picture vs. case-by-case scenario: An interesting finding is that the authors observe that in most cases, there is no substantial evidence for either the absence or the presence of an effect, as evidenced by the equivalence tests. Thus, using both suggested criteria results in a picture similar to the one initially raised by the paper itself. The work done by the authors highlights additional criteria that can be used to further analyze replication success on a case-by-case basis, and I believe that this is where the paper's main contributions lie. Despite not changing the overall picture much, I agree that the p-value criterion by itself does not distinguish between (1) a situation where the original study had low statistical power, resulting in a highly inconclusive non-significant result that does not provide evidence for the absence of an effect and (2) a scenario where the original study was adequately powered, and a non-significant result may indeed provide some evidence for the absence of an effect when analyzed with appropriate methods. Equivalence testing and Bayesian factor approaches are valuable tools in both cases.Regarding the 0.05 threshold, the choice of the prior distribution for the SMD under the alternative H1 is debatable, and this also applies to the equivalence margin. Sensitivity analyses, as highlighted by the authors, are helpful in these scenarios.

Thank you for the thorough review and constructive feedback. We have added an additional “Appendix C: Null results from the RPP and EPRP” that shows equivalence testing and Bayes factor analyses for the RPP and EPRP null results.

**Reviewer #3 (Public Review):**
Summary:The paper points out that non-significance in both the original study and a replication does not ensure that the studies provide evidence for the absence of an effect. Also, it can not be considered a "replication success". The main point of the paper is rather obvious. It may be that both studies are underpowered, in which case their non-significance does not prove anything. The absence of evidence is not evidence of absence! On the other hand, statistical significance is a confusing concept for many, so some extra clarification is always welcome.One might wonder if the problem that the paper addresses is really a big issue. The authors point to the "Reproducibility Project: Cancer Biology" (RPCB, Errington et al., 2021). They criticize Errington et al. because they "explicitly defined null results in both the original and the replication study as a criterion for replication success." This is true in a literal sense, but it is also a little bit uncharitable. Errington et al. assessed replication success of "null results" with respect to 5 criteria, just one of which was statistical (non-)significance.It is very hard to decide if a replication was "successful" or not. After all, the original significant result could have been a false positive, and the original null-result a false negative. In light of these difficulties, I found the paper of Errington et al. quite balanced and thoughtful. Replication has been called "the cornerstone of science" but it turns out that it's actually very difficult to define "replication success". I find the paper of Pawel, Heyard, Micheloud, and Held to be a useful addition to the discussion.Strengths:This is a clearly written paper that is a useful addition to the important discussion of what constitutes a successful replication.Weaknesses:To me, it seems rather obvious that non-significance in both the original study and a replication does not ensure that the studies provide evidence for the absence of an effect. I'm not sure how often this mistake is made.

Thanks for the feedback. We do not have systematic data on how often the mistake of confusing absence of evidence with evidence of absence has been made in the replication context, but we do know that it has been made in at least three prominent large-scale replication projects (the RPP, RPEP, RPCB). We therefore believe that there is a need for our article.

Moreover, we agree that the RPCB provided a nuanced assessment of replication success using five different criteria for the original null results. We emphasize this now more in the “Introduction” section. However, we do not consider our article as “a little bit uncharitable” to the RPCB, as we discuss all other criteria used in the RPCB and note that our intent is not to diminish the important contributions of the RPCB, but rather to build on their work and provide constructive recommendations for future researchers. Furthermore, in response to comments made by Reviewer #2, we have added an additional “Appendix B: Null results from the RPP and EPRP” that shows equivalence testing and Bayes factor analyses for null results from two other replication projects, where the same issue arises.

**Reviewer #1 (Recommendations For The Authors):**
The authors may wish to address the dichotomy issue I raise above, either in the analysis or in the discussion.

Thank you, we now emphasize that Bayes factors and TOST p-values do not need to be dichotomized but can be interpreted as quantitative measures of evidence, both in the “Methods for assessing replicability of null results” and the “Conclusions” sections.

**Reviewer #2 (Recommendations For The Authors):**
Given that, here follow additional suggestions that the authors should consider in light of the manuscript's word count limit, to avoid confusing the paper's main idea:1. Referencing: Could you reference the three interesting cases among the 15 RPCB null results (specifically, the three effects from the original paper #48) where the Bayes factor differs qualitatively from the equivalence test?

We now explicitly cite the original and replication study from paper #48.

1. Equivalence testing: As the authors state, only 4 out of the 15 study pairs are able to establish replication success at the 5% level, in the sense that both the original and the replication 90% confidence intervals fall within the equivalence range. Among these 4, two (Paper #48, Exp #2, Effect #5 and Paper #48, Exp #2, Effect #6) were initially positive with very low p-values, one (Paper #48, Exp #2, Effect #4) had an initial p of 0.06 and was very precisely estimated, and the only one in which equivalence testing provides a clearer picture of replication success is Paper #41, Exp #2, Effect #1, which had an initial p-value of 0.54 and a replication p-value of 0.05. In this latter case (or in all these ones), one might question whether the "liberal" equivalence range of Δ = 0.74 is the most appropriate. As the authors state, "The post-hoc specification of equivalence margins is controversial."

We agree that the post hoc choice of equivalence ranges is a controversial issue. The margins define an equivalence region where effect sizes are considered practically negligible, and we agree that in many contexts SMD = 0.74 is a large effect size that is not practically negligible. We therefore present sensitivity analyses for a wide range of margins. However, we do not think that the choice of this margin is more controversial for the mentioned studies with low p-values than for other studies with greater p-values, since the question of whether a margin plausibly encodes practically negligible effect sizes is not related to the observed p-value of a study. Nevertheless, for the new analyses of the RPP and EPRP data in Appendix B, we have added additional sensitivity analyses showing how the individual TOST p-values and Bayes factors vary as a function of the margin and the prior standard deviation. We think that these analyses provide readers with an even more transparent picture regarding the implications of the choice of these parameters than the “project-wise” sensitivity analyses in Appendix A.

1. Bayes factor suggestions: For the Bayes factor approach, it would be interesting to discuss examples where the BF differs slightly. This is likely to occur in scenarios where sample sizes differ significantly between the original study and replication. For example, in Paper #48, Exp #2 and Effect #4, the initial p is 0.06, but the BF is 8.1. In the replication, the BF dramatically drops to < 1/1000, as does the p-value. The initial evidence of 8.1 indicates some evidence for the absence of an effect, but not strong evidence ("strong evidence for H0"), whereas a p-value of 0.06 does not lead to such a conclusion; instead, it favors H1. It would be interesting if the authors discussed other similar cases in the paper. It's worth noting that in Paper #5, Exp #1, Effect #3, the replication p-value is 0.99, while the BF01 is 2.4, almost indicating "moderate" evidence for H0, even though the p-value is inconclusive.

We agree that some of the examples nicely illustrate conceptual differences between p-values and Bayes factors, e.g., how they take into account sample size and effect size. As methodologists, we find these aspects interesting ourselves, but we think that emphasizing them is beyond the scope of the paper and would distract eLife readers from the main messages.

Concerning the conceptual differences between Bayes factors and TOST p-values, we already discuss a case where there are qualitative differences in more detail (original paper #48). We added another discussion of this phenomenon in the Appendix C as it also occurs for the replication of Ranganath and Nosek (2008) that was part of the RPP.

1. p-values, magnitude and precision: It's noteworthy to emphasize, if the authors decide to discuss this, that the p-value is influenced by both the effect's magnitude and its precision, so in Paper #9, Exp #2, Effect #6, BF01 = 4.1 has a higher p-value than a BF01 = 2.3 in its replication. However, there are cases where both p-values and BF agree. For example, in Paper #15, Exp #2, Effect #2, both the original and replication studies have similar sample sizes, and as the p-value decreases from p = 0.95 to p = 0.23, BF01 decreases from 5.1 ("moderate evidence for H0") to 1.3 (region of "Absence of evidence"), moving away from H0 in both cases. This also occurs in Paper #24, Exp #3, Effect #6.

We appreciate the suggestions but, as explained before, think that the message of our paper is better understood without additional discussion of more general differences between p-values and Bayes factors.

1. The grey zone: Given the above topic, it is important to highlight that in the "Absence of evidence grey zone" for the null hypothesis, for example, in Paper #5, Exp #1, Effect #3 with a p = 0.99 and a BF01 = 2.4 in the replication, BF and p-values reach similar conclusions. It's interesting to note, as the authors emphasize, that Dawson et al. (2011), Exp #2, Effect #2 is an interesting example, as the p-value decreases, favoring H1, likely due to the effect's magnitude, even with a small sample size (n = 3 in both original and replications). Bayes factors are very close to one due to the small sample sizes, as discussed by the authors.

We appreciate the constructive comments. We think that the two examples from Dawson et al. (2011) and Goetz et al. (2011) already nicely illustrate absence of evidence and evidence of absence, respectively, and therefore decided not to discuss additional examples in detail, to avoid redundancy.

1. Using meta-analytical results (?): For papers from RPCB, comparing the initial study with the meta-analytical results using Bayes factor and equivalence testing approaches (thus, increasing the sample size of the analysis, but creating dependency of results since the initial study would affect the meta-analytical one) could change the conclusions. This would be interesting to explore in initial studies that are replicated by much larger ones, such as: Paper #9, Exp #2, Effect #6; Goetz et al. (2011), Exp #1, Effect #1; Paper #28, Exp #3, Effect #3; Paper #41, Exp #2, Effect #1; and Paper #47, Exp #1, Effect #5).

Thank you for the suggestion. We considered adding meta-analytic TOST p-values and Bayes factors before, but decided that Figure 3 and the results section are already quite technical, so adding more analyses may confuse more than help. Nevertheless, these meta-analytic approaches are discussed in the “Conclusions” section.

1. Other samples of fields of science: It would be interesting to investigate whether using Bayes factors and equivalence tests in addition to p-values results in a clearer scenario when applied to replication data from other fields. As mentioned by the authors, the Reproducibility Project: Experimental Philosophy (RPEP) and the Reproducibility Project: Psychology (RPP) have data attempting to replicate some original studies with null results. While the RPCB analysis yielded a similar picture when using both criteria, it is worth exploring whether this holds true for RPP and RPEP. Considerations for further research in this direction are suggested. Even if the original null results were excluded in the calculation of an overall replicability rate based on significance, sensitivity analyses considering them could have been conducted. The present authors can demonstrate replication success using the significance criteria in these two projects with initially p < 0.05 studies, both positive and non-positive.

Thank you for the excellent suggestion. We added an Appendix B where the null results from the RPP and EPRP are analyzed with our proposed approaches. The results are also discussed in the “Results” and “Conclusions” sections.

1. Other approaches: I am curious about the potential impact of using an approach based on equivalence testing (as described in https://arxiv.org/abs/2308.09112). It would be valuable if the authors could run such analyses or reference the mentioned work.

Thank you. We were unaware of this preprint. It seems related to the framework proposed by Stahel W. A. (2021) New relevance and significance measures to replace p-values. PLoS ONE 16(6): e0252991.https://doi.org/10.1371/journal.pone.0252991

We now cite both papers in the discussion.

1. Additional evidence: There is another study in which replications of initially p > 0.05 studies with p > 0.05 replications were also considered as replication successes. You can find it here: https://www.medrxiv.org/content/10.1101/2022.05.31.22275810v2. Although it involves a small sample of initially p > 0.05 studies with already large sample sizes, the work is currently under consideration for publication in PLOS ONE, and all data and materials can be accessed through OSF (links provided in the work).

Thank you for sharing this interesting study with us. We feel that it is beyond the scope of the paper to include further analyses as there are already analyses of the RPCB, RPP, and EPRP null results. However, we will keep this study in mind for future analysis, especially since all data are openly available.

1. Additional evidence 02: Ongoing replication projects, such as the Brazilian Reproducibility Initiative (BRI) and The Sports Replication Centre (https://ssreplicationcentre.com/), continue to generate valuable data. BRI is nearing completion of its results, and it promises interesting data for analyzing replication success using p-values, equivalence regions, and Bayes factor approaches.

We now cite these two initiatives as examples of ongoing replication projects in the introduction. Similarly as for your last point, we think that it is beyond the scope of the paper to include further analyses as there are already analyses of the RPCB, RPP, and EPRP null results.

**Reviewer #3 (Recommendations For The Authors):**
I have no specific recommendations for the authors.

Thank you for the constructive review.

**Reviewing Editor (Recommendations For the Authors):**
I recognize that it was suggested to the authors by the previous Reviewing Editor to reduce the amount of statistical material to be made more suitable for a non-statistical audience, and so what I am about to say contradicts advice you were given before. But, with this revised version, I actually found it difficult to understand the particulars of the construction of the Bayes Factors and would have appreciated a few more sentences on the underlying models that fed into the calculations. In my opinion, the provided citations (e.g., Dienes Z. 2014. Using Bayes to get the most out of non-significant results) did not provide sufficient background to warrant a lack of more technical presentation here.

Thank you for the feedback. We added a new “Appendix C: Technical details on Bayes factors” that provides technical details on the models, priors, and calculations underlying the Bayes factors.